# AdaptSim: Task-Driven Simulation Adaptation for Sim-to-Real Transfer

**Abstract:** Simulation parameter settings such as contact models and object geometry approximations are critical to training robust manipulation policies capable of transferring from simulation to real-world deployment. There is often an *irreducible gap* between simulation and reality: attempting to match the dynamics between simulation and reality may be infeasible and may not lead to policies that perform well in reality for a specific task. We propose AdaptSim, a new *task-driven* adaptation framework for sim-to-real transfer that aims to optimize task performance in target (real) environments. First, we meta-learn an adaptation policy in simulation using reinforcement learning for adjusting the simulation parameter distribution based on the current policy's performance in a target environment. We then perform iterative real-world adaptation by inferring new simulation parameter distributions for policy training. Our extensive simulation and hardware experiments demonstrate AdaptSim achieving 1-3x asymptotic performance and ~2x real data efficiency when adapting to different environments, compared to methods based on Sys-ID and directly training the task policy in target environments.

## 1 Introduction

Learning robust and generalizable policies for real-world manipulation tasks typically requires a substantial amount of training data. Since using real data exclusively can be very expensive or even infeasible, we often resort to training mostly in simulation. This raises the question: how should we specify simulation parameters to maximize performance in the real world while minimizing the amount of real-world data we require?

A popular method is to perform *domain randomization* [1, 2, 3, 4]: train a policy using a wide range of different simulation parameters in the hope that the policy can thus handle possible real-world variations in dynamics or observations. However, the trained policy may achieve good *average* performance, but perform poorly in a particular real environment. There has been work in performing system identification (Sys-ID) for providing a point or a distributional estimate of parameters that best matches the robot or environment dynamics exhibited in real-world data. This estimation can be performed using either a single iteration [5] or multiple ones [6]. These *adaptive* domain randomization techniques allow training policies suited to specific target environments.

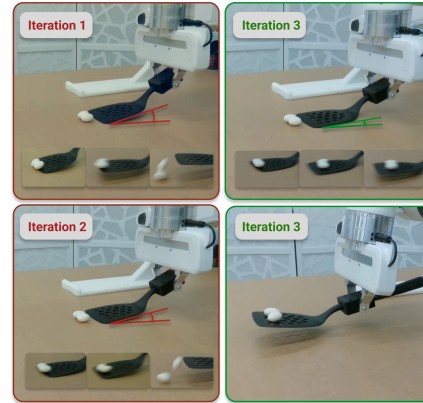

Figure 1: AdaptSim iteratively improves task performance in dynamic scooping task under "irreducible" sim-to-real gap.

While simple objects such as a box and its properties like the inertia can be well-modeled, there is a substantial amount of *"irreducible"* sim-to-real gap in many settings such as contact-rich manipulation tasks. Consider the task of using a cooking spatula to dynamically scoop up small pieces of food from a table (Fig. 1). The exact geometry of the pieces and spatula is difficult to specify, deformations such as the spatula bending against the table are not yet maturely implemented in simulators, and contact models such as point contact have been known to poorly approximate the complex real-world contact behavior [7] in these settings. In this

case, real environments are out-of-domain (OOD) from simulation, and performing Sys-ID of simulation parameters might fail to train a useful policy for the real world due to this inherent irreducible gap.

**Contributions.** In this work we take a *task-driven* approach: instead of trying to align the simulation with real-world dynamics, we focus on finding simulation parameters such that the resulting policy optimizes task performance. Such an approach can lead to policies that achieve high reward in the real world even with an irreducible sim-to-real gap. We consider settings where the robot has access to a simulator between iterations of real-world interactions, allowing it to observe real-world dynamics and adapt the simulator accordingly with the goal of improving task performance in reality. We propose *AdaptSim* — a two-phase framework where (i) an adaptation policy that updates simulation parameters is first meta-trained using reinforcement learning in simulation, and (ii) then deployed on the real environment iteratively. Training the adaptation policy to maximize task reward enhances the efficiency of real data usage by identifying only task-relevant simulation parameters and helps trained policies better generalize to OOD (real) environments. We demonstrate our approach achieving 1-3x asymptotic performance and $\sim$2x real data efficiency in OOD environments in three robotic tasks including two that involve contact-rich manipulation, compared to methods based on Sys-ID and directly training the task policy in target environments.

## 2 Related Work

Sim-to-real transfer in robotics has been primarily addressed using Domain Randomization (DR) techniques [1, 8, 9, 10, 11, 12, 13] that inject noise in simulation parameters related to visuals, dynamics, and actuations. Below we summarize techniques that better adapt to real environments.

**Sys-ID domain adaptation.** Inspired by classical work in Sys-ID [14, 15], there has been a popular line of work identifying simulation parameters that match the robot and environment dynamics in the real environment. BayesSim [6] and follow-up work [16, 17] apply Bayesian inference to iteratively search for a posterior distribution of the simulation parameters based on simulation and real-world trajectories. However, these methods consider relatively well-modeled environment parameterizations such as object mass or friction coefficient during planar contact; Sys-ID approaches can be brittle when the simulation does not closely approximate the real world [13, 18].

**Task-driven domain adaptation.** AdaptSim better fits within a different line of work that aims to find simulation parameters that maximize the task reward in target environments. Muratore et al. [19] apply Bayesian Optimization (BO) to optimize parameters such as pendulum pole mass and joint damping coefficient in a real pendulum swing-up task. Other work focus on adapting to simulated domains only [20, 21, 22]. One major drawback of these methods is that they require a large number of rollouts in target environments (e.g., 150 in [19]), which is very time-consuming for many tasks requiring human reset. AdaptSim meta-learns adaptation strategies in simulation and requires only a few real rollouts for inference (*e.g.,* 20 in our pushing experiments).

## 3 Problem Formulation

**Environment.** In simulation, we consider a space $\Omega$ that parameterizes quantities such as friction coefficients and dimensions of geometric primitives. Let $\mathcal{E}$ denote a distribution of sim parameters with support on $\Omega$. Denote a single sim environment $E \in \Omega$ and a real environment $E^r$.

**Task Policy and Trajectory.** We denote a task policy $\pi \in \Pi : \mathcal{O} \to \mathcal{A}$ that maps the robot's observation $o_t$ to action $a_t$. Running it in an environment results in a state-action trajectory $\tau(\pi; E) : [0,T] \times \Pi \times \mathcal{E} \to \mathcal{S} \times \mathcal{A}$ with time horizon $T$. The trajectory is also subject to an initial state distribution. We specify tasks for the robot using a reward function (*e.g.,* pushing some object to a specific location on the table), and let $R(\tau) \in [0,1]$ denote the normalized cumulative reward accrued by a trajectory. We let $R(\pi; E)$ denote the reward of running the task policy $\pi$ in the environment $E$, in expectation over the initial state distribution.

**Goal.** Our eventual goal is to find a task policy that maximizes the task performance in a real environment $E_r$. Instead of directly searching for the policy, we search for the best sim parameter distribution $\mathcal{E}$ for training $\pi$ in the following bi-level optimization objective:

$$\sup_{\mathcal{E}} R(\pi_{\mathcal{E}}^*; E^r), \text{ where } \pi_{\mathcal{E}}^* := \sup_{\pi} \mathbb{E}_{E \sim \mathcal{E}} [R(\pi; E)], \tag{1}$$

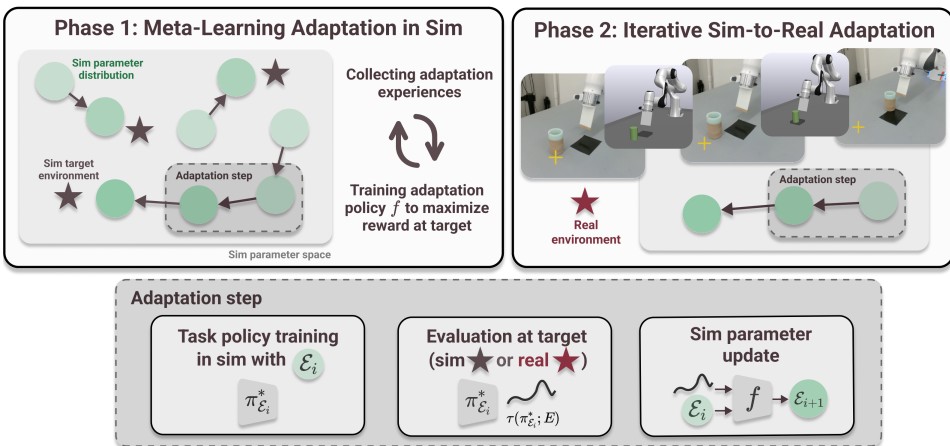

Figure 2: AdaptSim consists of two phases: (1) meta-training an adaptation policy in sim by maximizing task reward on randomly sampled simulated target environments; (2) iteratively adapting simulation parameter distributions based on real trajectories. The upper-right illustration shows that using only a few real trajectories, the task policy is adapted to push the bottle closer to the target location (yellow cross).

the optimal task policy for $\mathcal{E}$. Performing the outer level of (1) requires interactions with $E^r$ (the real world); we allow a small budget of such interactions. We emphasize that the objective above identifies the optimal distribution of simulation parameters for maximizing task performance, unlike objectives that attempt to match the dynamics between simulation and reality.

# 4 Approach

One way to solve (1) is to perform blackbox optimization on $\mathcal{E}$ by evaluating $R(\pi_{\mathcal{E}}^*;E^r)$ [19], which requires a large budget of real trajectories (see results in Sec. 6). AdaptSim instead amortizes the expensive outer loop to simulation: it solves (1) for many simulated environments, learns the mapping to the solutions, and then *infers* the solution for $E_r$. There are two phases (Fig. 2):

1) **Meta-learn the adaptation policy in sim**: randomly sample target environments $E^s \in \Omega$ in sim, and then train an "adaptation" policy $f:(\mathcal{E},\tau) \mapsto \Delta_{\mathcal{E}}$ using RL to maximize task reward in $E^s$, by updating the sim parameter distribution (and the corresponding task policies) in iterations.

2) **Iteratively adapt sim parameters with real data**: given a real environment $E^r$, iteratively infer better sim parameter distributions using the trained $f$ and a few real trajectories; the task policy is iteratively fine-tuned in sim to improve task reward with the updated parameter distribution.

## 4.1 Phase 1: meta-learning the adaptation policy in sim

In order to correctly infer simulation parameters for an unseen real environment at test time, we first train the adaptation policy to infer better parameters for many simulated target environments. This phase happens entirely in simulation. Formally, we model the problem as a partially-observable contextual bandit [23].

**Definition 1** *A Simulation-Adaptation Contextual Bandit (SA-CB) is specified by a tuple* $(\Omega,\mathcal{T},\mathcal{P},R)$:

- $\Omega$ *is the space of contexts. Each context corresponds to a simulated target environment* $E^s$; *the context is not directly observable.*
- $\mathcal{T}$ *is the space of partial observations of the context. Each observation corresponds to a trajectory observed by running the task policy in a given context.*
- $\mathcal{P}$ *is the space of actions. An action corresponds to choosing a sim parameter distribution* $\mathcal{E}$.
- $R$ *is the reward associated with choosing an action in a particular context* (i.e., *the reward* $R(\pi_{\mathcal{E}}^*;E^s)$ *of the task policy* $\pi_{\mathcal{E}}^*$ *trained with* $\mathcal{E}$ *when deployed in the target environment* $E^s$).

It may be difficult to infer the optimal $\mathcal{E} \in \mathcal{P}$ using a single iteration of interactions with the target environment — if the current task policy fails badly in the target environment, the interaction may reveal little information. Thus, we iteratively apply incremental changes to $\mathcal{E}$, with the parameter distribution initialized as $\mathcal{E}_{i=0}$. Solving the SA-CB (using techniques that we detail below), we meta-learn an adaptation

policy $f(\mathcal{E},\tau)$ to maximize:

$$\mathop{\mathbb{E}}_{E^s \sim \mathcal{U}_\Omega} \mathop{\mathbb{E}}_{\mathcal{E}_0 \sim \mathcal{U}_\mathcal{P}} \sum_{i=0}^{I} \gamma^i R(\pi^*_{\mathcal{E}_i};E^s), \tag{2}$$

$$\text{where } \mathcal{E}_{i+1} = \mathcal{E}_i + \Delta_{\mathcal{E}_i}, \Delta_{\mathcal{E}_i} = f\big(\mathcal{E}_i, \tau(\pi^*_{\mathcal{E}_i};E^s)\big),$$

and $\mathcal{U}_\Omega$ and $\mathcal{U}_\mathcal{P}$ are uniform distributions over $\Omega$ and $\mathcal{P}$ respectively, and $\gamma < 1$ is the discount factor. This is the expected discounted sum of task rewards over multiple interactions from $i = 0$ to the adaptation horizon $I$, over random sampling of simulated target environment and initial sim parameter distribution.

**Sim parameter distribution space.** We choose the space $\mathcal{P}$ of possible simulation parameter distributions to be Gaussian with mean bounded within $\Omega$ and a fixed variance. We also use a fixed step size $\delta$ for adapting each simulation parameter, ranging from $10\%$ to $15\%$ of the parameter range depending on the dimension of $\Omega$ — thus the set of possible $\Delta_\mathcal{E}$ along each dimension is $\{\delta, -\delta, 0\}$.

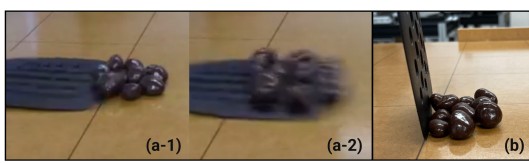

(a-1)  (a-2)  (b)

Figure 3: Task-policy trajectories better reveal task-relevant information such as scooping dynamics under fast contact.

---

**Algorithm 1** Meta-learning the adaptation policy in sim

---

**Require:** $(\Omega, \mathcal{T}, \mathcal{P}, R)$, SA-CB
**Require:** $S_f = \varnothing$, replay buffer
**Require:** $S_\mathcal{E} = \varnothing$, set of all simulation parameter distributions (and their task policies) used
1: Initialize $\epsilon \leftarrow 1$
2: **for** $k \leftarrow 0$ to $K$ **do**
3:     Sample target $E^s \sim \mathcal{U}_\Omega$ and $\mathcal{E}_{i=0} \sim \mathcal{U}_\mathcal{P}$
4:     **for** $i \leftarrow 0$ to $I$ **do**
5:         Train task policy $\pi^*_{\mathcal{E}_i}$ (Sec.A2.2)
6:         Collect $\tau(\pi^*_{\mathcal{E}_i};E^s)$ and $R(\pi^*_{\mathcal{E}_i};E^s)$
7:         Sample random $\Delta_{\mathcal{E}_i}$ or infer $\Delta_{\mathcal{E}_i} = f(\mathcal{E}_i, \tau(\cdot;\cdot))$
8:         Update $\mathcal{E}_{i+1} \leftarrow \mathcal{E}_i + \Delta_{\mathcal{E}_i}$
9:         Add $\big(\mathcal{E}_i, \Delta_{\mathcal{E}_i}, \tau(\cdot;\cdot), R(\cdot;\cdot)\big)$ to $S_f$
10:        Add $\mathcal{E}_i$ (and $\pi^*_{\mathcal{E}_i}$) to $S_\mathcal{E}$
11:    **end for**
12:    Train $f$ using Double Q-Learning and $S_f$
13:    Anneal $\epsilon$ towards 0
14: **end for**
15: **return** $f, S_\mathcal{E}$

---

**Task-policy trajectory as observation.** We have chosen the task policy $\pi^*_\mathcal{E}$ to generate the trajectory observations used by the adaptation policy. Our intuition is that, compared to arbitrary policies or ones that generate the most "informative" trajectories in terms of dynamics [24], $\pi^*_\mathcal{E}$ better reveal the task-relevant information of the target environment . In the scooping task, the robot needs to attempt to scoop up the pieces so it can learn about the effect of the environment on the task (e.g., a piece with a flat bottom is generally harder to scoop). Simply pushing the pieces around does not exhibit the behavior of the pieces under fast contact (Fig. 3).

**Training the adaptation policy using RL.** The adaptation policy $f$ is parameterized using a Branching Dueling Q-Network [25], which outputs the state-action value of choosing any of the $\{\delta, -\delta, 0\}$ along each action dimension. It takes in (1) the vector of the mean of current simulation parameter distribution and (2) trajectory observation. We apply reinforcement learning (RL) to train $f$ to maximize Eq. (2). In simulation, we collect $K$ "adaptation trajectories"; each trajectory is a set $\{\big(\mathcal{E}_i, \Delta_{\mathcal{E}_i}, R(\pi^*_{\mathcal{E}_i};E^s), \tau(\pi^*_{\mathcal{E}_i};E^s)\big)\}_{i=0}^{I}$ and saved in a replay buffer $S_f$. Since each step involves training the corresponding task policy $\pi^*_{\mathcal{E}_i}$, which can be expensive, we apply off-policy Double Q-Learning [26] for sample efficiency. Using this, the adaptation policy outputs the greedy action of a parameterized Q function, $f(\mathcal{E},\tau) = \arg\max_{\Delta_\mathcal{E}} Q(\mathcal{E},\tau;\Delta_\mathcal{E})$. We use $\epsilon$-greedy exploration with $\epsilon$ initialized at 1 and annealed to 0.

This constitutes the first phase of AdaptSim. Algorithm 1 details the steps for collecting adaptation trajectories in the inner loop (Line 4-15) and meta-learning the adaptation policy. We save all distributions (and their corresponding task policies, omitted in notations for convenience) in a set $S_\mathcal{E}$, which are used again in the second phase. Training the task policy for each $\mathcal{E}$ is the most computationally heavy component of Algorithm 1; in Sec. A2.2 we explain the heuristics applied to allow re-using task policies between $\mathcal{E}$ in order to improve computational efficiency.

### 4.2   Phase 2: iteratively adapt sim parameters with real data

After meta-training the adaptation policy to find good task policies for a diverse set of target environments in simulation, we can apply it for inference and perform adaptation for the real environment $E^r$. Algorithm 2 details the iterative process. We apply the same adaptation process as the inner loop of Algorithm 1 for $I^r$ iterations: train the task policy in simulation, evaluate it in the real environment, and infer the change of simulation parameters based on real trajectories. We always apply the greedy action from $f(\mathcal{E},\tau)$ ($\epsilon = 0$).

**Algorithm 2** Iteratively adapt sim parameters with real data

**Require:** $E^r$, real environment
**Require:** $(\Omega, \mathcal{T}, \mathcal{P}, R)$, SA-CB
**Require:** $f$, adaptation policy trained in Phase 1
**Require:** $S_f$, set of sim parameter distributions (and corresponding task policies) from Phase 1

1: Sample $S'_f$ from $S_f$
2: **for** $i \leftarrow 0$ to $I^r$ **do**
3:     **for** $\mathcal{E}_i \in S'_f$ **do**
4:         Train or fine-tune the task policy $\pi^*_{\mathcal{E}_i}$ in sim
5:         Collect $\tau(\pi^*_{\mathcal{E}_i}; E^r)$ and $R(\pi^*_{\mathcal{E}_i}; E^r)$ in real
6:         Update $\mathcal{E}_{i+1} \leftarrow \mathcal{E}_i + f\big(\mathcal{E}_i, \tau(\cdot; \cdot)\big)$
7:     **end for**
8: **end for**
9: **return** $\pi^*_{\mathcal{E}_i}$ with the highest $R(\cdot; E^r)$

Since we have sampled a large set $S_{\mathcal{E}}$ of parameter distributions and trained their task policies in Phase 1, we may re-use them here. At the beginning of Phase 2, we sample $S'_{\mathcal{E}}$, a set of $N$ distributions saved in $S_{\mathcal{E}}$, as the initial distributions to be adapted independently. Usually we pick $N = 2$ considering the trade-off between number of real trajectories needed and convergence of task performance (see Appendix A5 for analysis).

## 5 Tasks

Next we detail the three robotic tasks for evaluating AdaptSim and baselines. We choose these tasks and design the environments to highlight the irreducible gap between training and test domains.

### 5.1 Swing-up of a linearized double pendulum

This is a classic control task where the goal is to swing up a simple double pendulum with two actuated joints at one end of the two links. We consider the dynamics linearized around the state with the pendulum at the top, and thus the optimal policy can be solved exactly using Linear Quadratic Regulator (LQR) [27] for a particular set of simulation parameters (*i.e.,* a Dirac delta distribution). The task cost (reward) function is defined with the standard quadratic state error and actuation penalty. The trajectory observation is evenly spaced points along trajectory of the two joints.

**Simulation setup.** The environment is parameterized with four parameters: $m_1$ and $m_2 \in [1,2]$, point mass of the two joints, and $b_1$ and $b_2 \in [1,2]$, damping coefficients of the two joints. The dynamics is simulated with numerical integration without a dedicated physics simulator.

### 5.2 Dynamic table-top pushing of a bottle

The robot needs to dynamically push a bottle to a particular target location on the table (Fig. 4). Since the target can be outside the workspace of the robot, the robot must push objects with high velocity — causing them to slide after a short period of contact. The task policy is parameterized with a neural network that maps the desired target location to action including (1) planar pushing angle and (2) robot end-effector speed (see Appendix A4 for visualization) and the predicted reward. The network then acts as a state-value (Q) function and is trained off-policy while simulated trajectories are saved in a replay buffer. The task cost (reward) is defined as the distance between the target location and the final location of the bottle. The trajectory observation is either (1) the final 2D position of the bottle only, or (2) evenly spaced points along the 2D trajectory — we consider both representations in the experiments.

**Simulation setup.** We employ the Drake physics simulator [28] for its accurate contact mechanics. In this simulated environment, a small patch of the table is simulated with different physics properties, simulating a wet or sticky area on the work surface. Parameter settings for this task are shown in Table 1. The hydroelastic modulus is a parameter of the hydroelastic contact model [7]

| Notation | Description | Range |
|---|---|---|
| $\mu$ | table friction coefficient | $[0.05, 0.2]$ |
| $e$ | hydroelastic modulus [7] | $[1e4, 1e6]$ |
| $\mu_p$ | patch friction coefficient | $[0.20, 0.80]$ |
| $y_p$ | patch lateral location | $[-0.10, 0.10]$ |

Table 1: Sim setup for the pushing task.

implemented in Drake — it roughly simulates how "soft" the contact is between the objects, with lower values being softer.

**Real setup.** Two 3D-printed bottles (Fig. 4, Heavy and Light) with the same dimensions but different materials and masses are used. With an idealized model, the sliding distance should only depend on the contact surface but not the mass — which is the case in simulation — but in real experiments, we find the two bottles consistently travel different distances. Additionally, Heavy tends to rotate slightly despite being pushed straight due to a slightly uneven bottom surface. This type of unmodeled effect exemplifies the irreducible sim-to-real gap. We also adhere a small piece of high-friction Neoprene rubber to the table, which decelerates the bottle and further complicate the task dynamics.

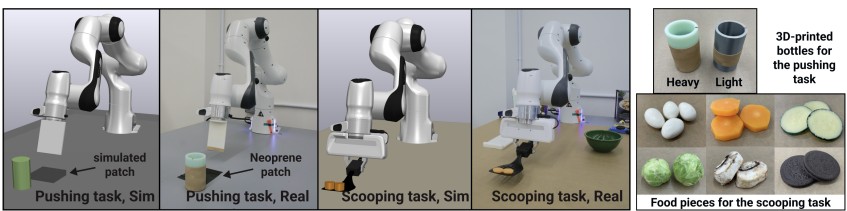

Figure 4: Setup of the dynamic pushing and dynamic scooping tasks in both simulation and reality.

### 5.3 Dynamic scooping of food pieces with a spatula

The robot needs to use a cooking spatula to scoop up small food pieces on the table (Fig. 4). It is a challenging task that requires intricate planning of the scooping trajectory — we notice humans cannot complete the task consistently without a few trials to practice. The task policy is parameterized with a neural network that maps the initial positions of the food pieces to parameterization of the scooping trajectory: (1) initial distance of the spatula from the pieces, (2) initial pitch angle of the spatula from the table, and (3) the timestep to lift up the spatula (see Appendix A4 for details), and the predicted reward. The task reward is defined as the ratio of food pieces on the spatula at the end of the action. The trajectory observation is evenly spaced points along 2D trajectories of the food pieces.

**Simulation setup.** We again use the Drake simulator. The parameter settings are shown in Table 2.

**Real setup.** Six different kinds of food pieces are used (Fig. 4): (1) chocolate raisin, (2) (fake, rubber-like) sliced carrot, (3) (fake, rigid) sliced cucumber, (4) raw Brussels sprout, (5) raw sliced mushroom, and (6) Oreo cookie. They cover different shapes from being round, ellipsoidal, to roughly cylindrical, and also have different amounts of deformation and friction.

| Notation | Description | Range |
|---|---|---|
| $\mu$ | friction coefficient | [0.25,0.4] |
| $e$ | hydroelastic modulus | [1e4,5e5] |
| $g$ | food piece geometry | {ellipsoid,cylinder} |
| $h$ | food piece height | [1.5cm,2.5cm] |

Table 2: Sim setup for the scooping task.

## 6 Experiments

Through extensive experiments below, we demonstrate that AdaptSim improves asymptotic task performance compared to Sys-ID and other baselines when adapting to real and OOD simulated environments, while also improving data efficiency. For baselines, first we consider methods that directly optimizes the task policy: (1) **Uniform domain randomization (UDR):** train a task policy to optimize the average task reward over environments from $\mathcal{U}_\Omega$; (2) **UDR+Target:** fine-tune the task policy from UDR with real data; (3) **LearnInTarget:** directly train a task policy with data in the target environment only by fitting a small neural network that maps action to final reward. The policy then outputs the action with the highest predicted reward. With enough real data, this baseline should act as the oracle or upper bound of task performance, but can be inefficient. Next, we consider two that perform SysID and iteratively train the task policy like AdaptSim: (4) **SysID-Bayes [6, 29]:** iteratively infer the sim parameter distribution based on real trajectories to match dynamics in sim and reality, known as BayesSim; (5) **SysID-Point:** infer a point estimate of the sim parameter instead of a distributional one (we hypothesize that in some settings randomizing sim parameters with a distribution can negatively impact task policy training).

### 6.1 AdaptSim achieves better task performance through adaptation

**Sim-to-Sim Adaptation.** We perform experiments for all baselines adapting to different WD (Within-Domain) and OOD simulated environments. WD environments are generated by sampling all simulation parameters within $\Omega$ of each task, and OOD environments are generated by sampling some parameters outside $\Omega$ (see Appendix A5 for details). Table 3 shows the adaptation results in the target environments in the three tasks. While Sys-ID baselines achieve high reward in WD environments, AdaptSim outperforms Sys-ID baselines in almost all OOD environments.

**Sim-to-Real Adaptation.** Next we perform experiments for adapting to real environments. Fig. 5 shows the average reward achieved at each adaptation iteration in the pushing and scooping tasks. Generally the performance gap between AdaptSim and Sys-ID baselines is larger in reality, with AdaptSim achieving better performance. In the scooping task, for example, AdaptSim is able to train a task policy for sliced cucumbers with decent performance (60% success rate); the pieces are very thin and difficult to scoop under (Fig. 8). Other baselines fail to scoop up the pieces.

| | Double Pendulum Swing-Up | | | | | Bottle Pushing | | | | | Food Scooping | | | | |
|---|---|---|---|---|---|---|---|---|---|---|---|---|---|---|---|
| Method | WD | OOD-1 | OOD-2 | OOD-3 | OOD-4 | WD | OOD-1 | OOD-2 | OOD-3 | OOD-4 | WD | OOD-1 | OOD-2 | OOD-3 | OOD-4 |
| AdaptSim | **0.98** | **0.96** | **0.95** | **0.95** | **0.98** | 0.95 | 0.87 | **0.73** | 0.86 | 0.77 | **1.00** | 0.64 | **1.00** | **1.00** | **0.55** |
| SysID-Bayes [6] | 0.85 | 0.76 | 0.79 | 0.23 | 0.96 | **0.98** | 0.80 | 0.65 | 0.81 | **0.79** | 0.90 | **0.66** | 0.81 | **1.00** | 0.36 |
| SysID-Point | 0.95 | 0.60 | 0.73 | 0.39 | 0.76 | 0.94 | 0.84 | 0.68 | 0.85 | 0.78 | 0.94 | 0.63 | 0.90 | **1.00** | 0.42 |
| UDR | - | - | - | - | - | 0.68 | 0.65 | 0.61 | 0.67 | 0.58 | 0.65 | 0.22 | 0.43 | 0.55 | 0.12 |
| UDR+Target | - | - | - | - | - | 0.78 | 0.73 | 0.66 | 0.71 | 0.70 | 0.61 | 0.31 | 0.49 | 0.60 | 0.21 |
| LearnInTarget | - | - | - | - | - | 0.91 | 0.75 | 0.66 | 0.74 | 0.71 | 0.03 | 0.00 | 0.25 | 0.26 | 0.03 |

Table 3: **Sim-to-Sim Adaptation.** Best average reward achieved over adaptation horizons at different WD and OOD simulated target environment in the three tasks. For the pendulum task, the values are normalized in [0,1] using the reward achieved by UDR (lower bound) and by using the best possible parameters within Ω (upper bound, estimated with exhaustive sampling). For the pushing task, the values are normalized with 20cm as the maximum error, which is the range of possible goal locations in the forward direction.

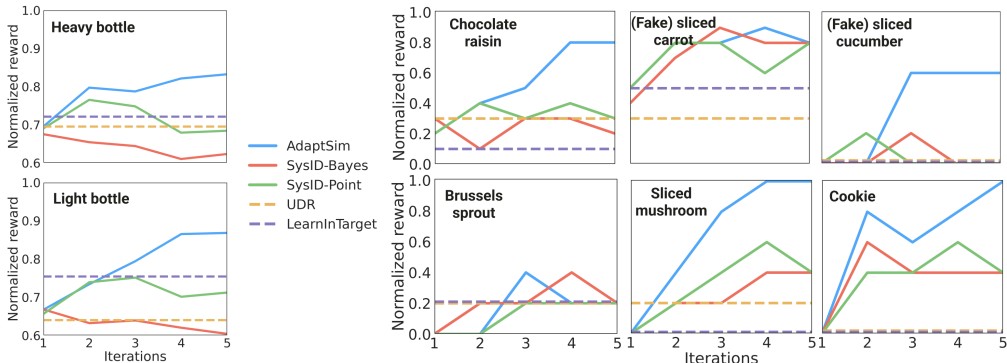

Figure 5: **Sim-to-Real Adaptation.** Reward achieved over adaptation iterations by all methods, in the task of pushing (left) and scooping up (right) different real objects (see Fig. 4 for images). Results are averaged over 10 trials in the pushing task and 5 in the scooping task.

### 6.2 AdaptSim improves real data efficiency

**Pushing task.** We compare AdaptSim with Learn-InTarget and UDR+Target with different number of real data budget. With enough data, LearnInTarget and UDR+Target should achieve high reward in the target environment. We do not compare with Sys-ID baselines here since Sec. 6.1 shows they typically fail to achieve the same level of task performance in real environments. In the task of pushing Heavy bottle, Table 4 shows that AdaptSim achieves a similar level of task performance (∼0.83) using only 16 trials while LearnInTarget and UDR+Target uses 40. Fine-tuning with real data in UDR+Target is ineffective until the real budget is sufficient and can negatively impact the performance in the low-data regime (*e.g.,* 4 and 8). This also exemplifies using simulation to amortize data requirements for policy training. We also introduce a new baseline **BayesOpt** here based on [19] that directly optimizes Eq. (1) with Bayesian Optimization. However, with 24 rollouts (the minimum needed to initialize the optimization) it only achieves 0.65.

| | Real data budget | | | | | | | |
|---|---|---|---|---|---|---|---|---|
| Method | 0 | 4 | 8 | 16 | 24 | 32 | 40 | 48 |
| AdaptSim | 0.30 | 0.69 | 0.80 | 0.83 | 0.84 | 0.84 | 0.82 | 0.83 |
| LearnInTarget | 0.05 | 0.04 | 0.63 | 0.69 | 0.76 | 0.80 | 0.84 | 0.83 |
| UDR+Target | 0.63 | 0.56 | 0.62 | 0.66 | 0.68 | 0.74 | 0.82 | 0.82 |
| BayesOpt | - | - | - | - | 0.65 | 0.72 | 0.79 | 0.80 |

Table 4: **Adaptation Data Efficiency.** Normalized reward achieved using different amount of real data in the pushing task with Heavy bottle.

**Larger improvement in scooping task.** While LearnInTarget and UDR+Target achieve reasonable performance in the pushing task, LearnInTarget achieves low reward on all the food pieces in the scooping task, and UDR+Target does not improve upon the performance of UDR policies. The action space in the scooping task is more complex and requires significantly more data to search for or improve task policies. AdaptSim's adaptation pre-training in simulation considerably amortizes the real data requirement.

### 6.3 AdaptSim finds sim parameters that are different from ones from SysID

We expect that AdaptSim finds simulation settings that achieve better task performance while not necessarily minimizing the full dynamics discrepancies between sim and reality. Fig. 6 shows SysID-Bayes finds parameters that are closer to the target in the parameter space, but for the pendulum task, such parameters lead to inferior task reward compared to those found by AdaptSim. Moreover, we compute the dynamics discrepancy, measured as the total variations between trajectories in the target environment and in the

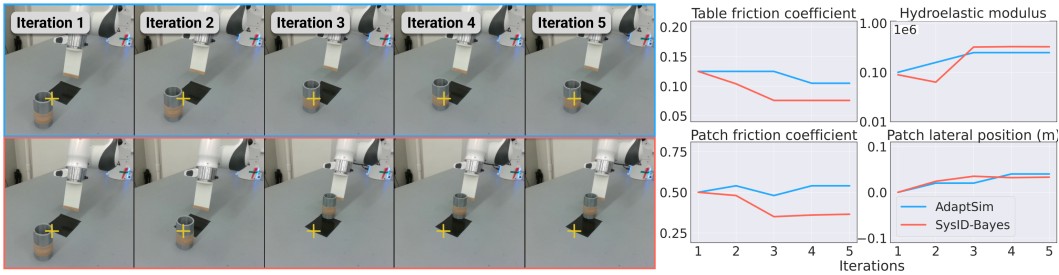

Figure 7: **Adaptation in Pushing Task.** AdaptSim correctly learns to push the bottle swiftly and close to the target. The task-relevant sim parameters learned by AdaptSim noticeably differ from those by SysID-Bayes which tends to underestimate table and patch friction, resulting in a less forceful push of the bottle and worse task performance.

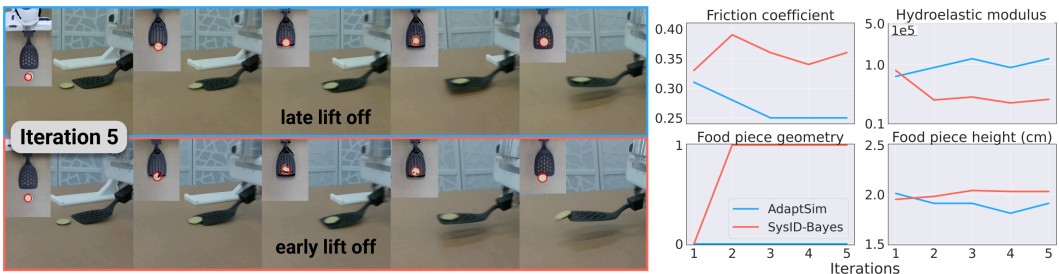

Figure 8: **Adaptation in Scooping Task.** With AdaptSim, the cucumber is successfully scooped up by lifting up the spatula off the table late; otherwise, the piece slips off the spatula. AdaptSim infers an ellipsoidal shape ($g = 1$, food piece geometry), while SysID-Bayes infers a cylindrical shape.

environment with adapted parameters. The results are 17.6 vs. 12.1 for AdaptSim and SysID-Bayes in OOD-1 environment, 21.7 and 11.1 in OOD-2, 39.9 and 16.4 in OOD-3, 75.8 and 56.4 in OOD-4. Thus for all four OOD target environments, SysID-Bayes finds sim parameters whose resulting dynamics are closer to the target environment (lower discrepancies), but Table 3 shows the task performance is worse.

Fig. 7 and Fig. 8 further show cases where SysID-Bayes under-performs AdaptSim and there are visible differences between sim parameter distributions found by the two approaches. In the pushing task, SysID-Bayes infers table and patch friction coefficients that are too low, and the trained task policy pushes the bottle with little speed. In the scooping task, interestingly, AdaptSim infers an ellipsoidal shape for the sliced cucumber despite it resembling a very thin cylinder, and the task policy achieves 60% success rate. Sys-ID infers a cylindrical shape but the task policy fails completely.

# 7    Discussions

**Summary.** We present *AdaptSim*, a framework for efficiently adapting simulation-trained task policies to the real world. AdaptSim meta-learns how to adapt simulation parameter distributions for better performance in diverse simulated target environments, and then infers better distributions for training real-world task policies using a small amount of real data.

**Limitations and Future Work.** In some settings AdaptSim does not outperform baselines (*e.g.,* OOD-4 in the pushing task and scooping up Brussels sprout in hardware, Fig. A8). First, AdaptSim's task-driven

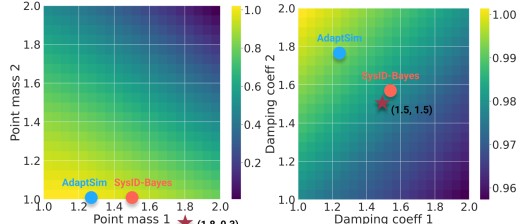

Figure 6: Sim parameters found by AdaptSim vs. SysID-Bayes in the OOD-1 setting of the double pendulum task. The colors indicate the maximum possible reward at each parameter. SysID-Bayes finds parameters closer to the target in the parameter space (dark red star), but the task performance is worse.

adaptation training requires the trained task policy being (nearly) optimal on the corresponding simulation parameter distribution — while it can be solved exactly in the double pendulum task, the task policy training in the two manipulation tasks can be noisy. Second, if the target environment is extremely OOD from the simulation domain and the adaptation policy has not been trained with similar trajectories, AdaptSim may not work as well. We believe the first issue can be mitigated by allowing more simulation budget for task policy training and better design of task policy re-use. The second issue can be addressed by designing the simulation parameter space $\Omega$ to better cover possible real-world behavior.

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

# Appendix

## A1 Extended Related Work

**Sys-ID domain adaptation.** Inspired by classical work in Sys-ID [14, 15], there has been a popular line of work identifying simulation parameters that match the robot and environment dynamics in the real environment before task policy training. BayesSim [6] and follow-up work [16, 17] applies Bayesian inference to iteratively search for a posterior distribution of the simulation parameters based on simulation and real-world trajectories. The inference problem has also been formulated using RL to minimize trajectory discrepancies [30]. A different approach [31, 32, 33] learns a residual model of dynamics (often parameterized with a neural network) to match simulation or an ideal physics model with reality. However, all these methods consider relatively well-modeled environment parameterizations such as object mass or friction coefficient during planar contact; Sys-ID approaches have been shown to fail in cases where the simulation does not closely approximate the real world [13, 18]. There is also work that avoids inferring the full dynamics but adapts with a low-dimensional latent representation online [34, 35, 36], but the representation is still trained with regression to match dynamics or simulation parameters. Importantly, the Sys-ID approaches highlighted above are all task-agnostic; this can lead to poor performance when trained task policies are sensitive to mismatches in dynamics between simulation and reality. Chi et al. [18] address the issue by using simulation to predict changes to trajectories from changes in actions as an implicit policy, but it requires the environment to be resettable, while AdaptSim works with randomly initialized object states.

**Task-driven domain adaptation.** AdaptSim better fits within a different line of work that aims to find simulation parameters that maximize the task reward in target environments. Muratore et al. [19] apply Bayesian Optimization (BO) to optimize parameters such as pendulum pole mass and joint damping coefficient in a real pendulum swing-up task. Other work focus on adapting to simulated domains only [20, 21, 22]. One major drawback of these methods is that they require a large number of rollouts in target environments (e.g., 700 in [19]), which is very time-consuming for many tasks requiring human reset. AdaptSim meta-learns adaptation strategies in simulation and requires only a few real rollouts for inference (*e.g.,* 20 in our pushing experiments). Liang et al. [37] apply the same task-driven objective to learn an exploration policy in manipulation tasks, but the task policy is synthesized using estimated simulation parameters via Sys-ID. Jin et al. [38] applies task-driven reduced-order model for dexterous manipulation tasks, but again the model is identified with Sys-ID and no vision-based control is involved. Ren et al. [39] search for adversarial environments (*e.g.,* objects) given the current task performance to robustify the policy, but unlike AdaptSim, the adversarial metric is measured in simulated domain only without real data.

**Learn to search/optimize.** Our work involves learning optimization strategies through meta-learning across a distribution of relevant problems, allowing for customization to the specific setting and increased sample efficiency [40, 41]. Chen et al. [42] meta-learns an RNN optimizer for black-box optimization. Volpp et al. [43] meta-learns the acquisition function in BO with RL; it is able to learn new exploration strategies for black-box optimization and tuning controller gains in sim-to-real transfer. Meta RL trains the task policy directly to optimize performance in new environments [44, 45, 46] — AdaptSim applies meta RL to optimize simulation parameters instead.

## A2 Additional details on approach

### A2.1 Sparse adaptation reward

In practice, we are only concerned with the reward if it reaches some minimum threshold — a bad task policy is not useful. Thus we use a sparse-reward version of Eq. (2),

$$\mathbb{E}_{E^s \sim \mathcal{U}_\Omega} \mathbb{E}_{\mathcal{E}_0 \sim \mathcal{U}_\mathcal{P}} \sum_{i=0}^{I} \gamma^i \mathbb{1}\left(R(\pi^*_{\mathcal{E}_i}; E^s) \geq \overline{R}\right) R(\pi^*_{\mathcal{E}_i}; E^s), \tag{A1}$$

where $\mathbb{1}()$ is the indicator function and $\overline{R}$ is the sparse-reward threshold. Using a sparse reward also discourages the adaptation policy from being myopic and getting trapped at a sub-optimal solution,

especially since we use a relatively small $I$ (*e.g.,* 5-10) in order to minimize the amount of real data, and use a small discount factor $\gamma$ (=0.9).

## A2.2  Task policy reuse across parameter distributions

Algorithm 1 requires training the task policy for each $\mathcal{E}$, which can be expensive with the two manipulation tasks. Our intuition is that we can share the task policy between parameter distributions of close distance, with the following heuristics:

- Record the total budget (*i.e.,* number of trajectories), and $j$, the number of simulation parameter distributions that a task policy has been trained with.

- Define distance between two parameter distribution $D(\cdot,\cdot)$ such as L2 distance between the mean. If $\mathcal{E}_i$ is within a threshold $\overline{D}$ from a previously seen distribution, re-use the task policy. If the policy is already trained with $M_{\max}$ budget total, do not train again; otherwise train with $\max(M_{\min}, \alpha^{j-1}M)$ budget, where $\alpha < 1$ and $M$ is the budget for training the policy for the first time.

- If the nearby parameter distribution re-uses a task policy, do not re-use the same policy again. This prevents the same task policy being used for too many $\mathcal{E}$.

**Remark 1** *re-using task policies between parameter distributions makes the reward $R$ depend on the adaptation history, as $\pi_{\mathcal{E}}^{*}$ depends on previous $\mathcal{E}$ that are used for training. We choose not to model this history dependency in $f$, as the reward should be largely dominated by the current $\mathcal{E}$.*

## A3  Additional details of adaptation policies

**Hyperparameters.** Table A1 shows the hyperparameters used for the adaptation policy training in Phase 1, including those defining the heuristics for re-using task policies among simulation parameter distributions. We generally use smaller adaptation step $\delta$ for smaller dimensional $\Omega$.

| | Task | | |
|---|---|---|---|
| Parameter | Pendulum | Pushing | Scooping |
| Total adaptation steps, $K$ | 1e4 | 1e4 | 1e4 |
| Adaptation horizon, $I$ | 10 | 8 | 8 |
| Adaptation step size, $\delta$ | 0.10 | 0.15 | 0.15 |
| Adaptation discount factor, $\gamma$ | 0.9 | 0.9 | 0.9 |
| Sprase reward threshold, $\overline{R}$ | 0.95 | 0.8 | 0.5 |
| Task policy reuse threshold, $\overline{D}$ | - | 0.16 | 0.16 |
| Task policy max budget, $M_{\max}$ | - | 3e4 | 4e3 |
| Task policy budget discount, $\alpha$ | - | 0.9 | 0.9 |
| Task policy init budget, $M$ | - | 1e4 | 1.2e3 |

Table A1: Hyperparameters used in adaptation policy training for the three tasks.

**Trajectory observations.** We detail the trajectory observation (as input to the adaptation policy) used in the three tasks.

- Pendulum task: each trial is 2.5 seconds long, and we use 12 evenly spaced points along the trajectories of the two joints, and thus each trajectory is 24 dimensional. For AdaptSim-State, SysID-Bayes-State, and SysID-Bayes-Point, again 12 points are used but sampled from the last 0.5 second only. One trajectory is used at each adaptation iteration — the trajectory input to the adaptation policy is 24 dimensional.

- Pushing task: each trial is 1.3 seconds long, and we use 6 evenly spaced points along the X-Y trajectory of the bottle, and thus each trajectory is also 12 dimensional. For AdaptSim-State, SysID-Bayes-State, and SysID-Bayes-Point, only the final X-Y position of the bottle is used. Two trajectories are used at each adaptation iteration — the trajectory input to the adaptation policy is 24 dimensional.

- Scooping task: each trial is 1 second long, and we use X-Y position of the food piece at the time step [0,0.2,0.3,0.4,0.5,0.6,0.8,1.0]s (more sampling around the initial contact between the spatula and the piece), and thus each trajectory is 16 dimensional. Two trajectories are used at each adaptation iteration — the trajectory input to the adaptation policy is 32 dimensional.

In real experiments, we track the bottle position in the pushing task using 3D point cloud information from a Azure Kinect RGB-D camera, which we find accurate. In the scooping task, the food pieces are too small and thin to be reliably tracked with point cloud, and thus we resort to extracting the contours from the RGB image and then finding the corresponding depth values at the same pixels in the depth image. During fast contact there can be motion blur around the food piece, and thus we add Gaussian noise with 0.2cm mean for X position and zero mean for Y position, and 0.2cm covariance for both, to the points in the ground-truth trajectories in simulation. We use positive mean in X since the motion blur tends to occur in the forward direction.

## A4    Additional details of the task setup and task policies

**Trajectory observation** First, we remove the action sequence from the task-policy trajectory and keep the state sequence only. Since the dynamics in real environments can be OOD, in order to achieve similar high-reward states as in simulated environments, the robot would need to use some actions not seen during training (or not seen for the particular state), hindering the adaptation policy to generalize if action sequence were included in the task policy trajectory. We assume that the task-relevant *state* sequence is covered by $\mathcal{T}$ if the task policy performs reasonably well in the real environment. This choice is also present in the state-only inverse RL literature [47] that addresses train-test dynamics mismatch. See Fig. A4 and related discussions in Sec. 6.3.

### A4.1    Dynamic pushing of a bottle

**Trajectory parameterization.** Here we detail the trajectory of the end-effector pusher designed for the task (Fig. A1). The trajectory is parameterized with two parameters: (1) planar pushing angle, which is the yaw orientation of the pusher relative to the forward direction that controls the direction of the bottle being pushed, and (2) forward speed (of the end-effector), in the direction specified by the pushing angle. The pushing angle varies between $-0.3$rad and $0.3$rad, and the forward speed varies between $0.4$m/s and $0.8$m/s. We find $0.8$m/s roughly the upper speed limit of the Franka Panda arm used. The pusher also pitches upwards during the motion and the speed is fixed to $0.8$rad/s. We design such trajectories to maximize the pushing distance at the hardware limit.

**Initial and goal states.** The bottle is placed at the fixed location ($x=0.56$m,$y=0$, relative to the arm base) on the table before the trial starts. The goal location is sampled from a region where the X location is between 0.7 and 1.0m and Y location is at most 10 degrees off from the centerline (Fig. A1 top-right). The patch, a 10cm by 10cm square, is placed at $x=0.75$m with its center (lateral position is varied as one of the simulation parameter).

**Task policy parameterization.** The task policy is parameterized using a Normalized Advantage Function (NAF) [48] that allows efficient Q Learning with continuous action output by restricting the Q value as a quadratic function of the action, and thus the action that maximizes the Q value can be found exactly without sampling. In this task, it maps the desired 2D goal location of the bottle to the two action parameters, planar pushing angle and forward speed. The policy is open-loop — the actions are determined before the trial starts and there is no feedback using camera observations.

**Hardware setup.** A 3D-printed, plate-like pusher is mounted at the end-effector instead of the paralle-jaw gripper in both simulation and reality. We also wrap elastic rubber bands around the bottom of the pusher and contact regions of the bottle to induce more elastic collision, which we find increases the sliding distance of the bottle.

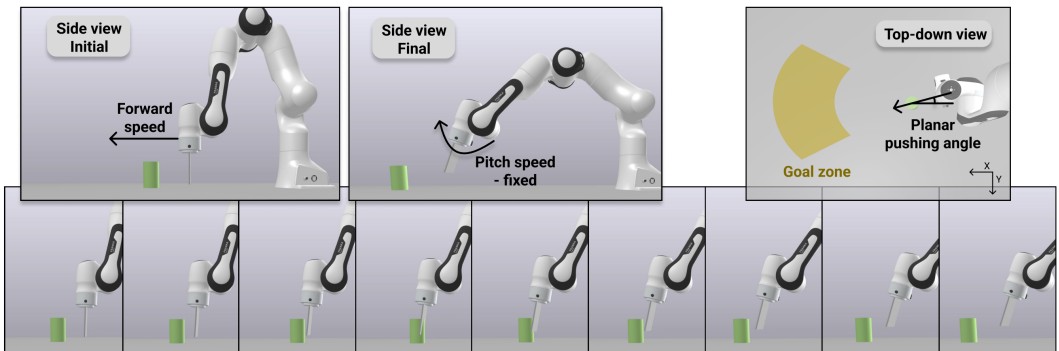

Figure A1: Visualization of the pushing trajectory and goal locations in the Drake simulator. There are two action parameters: (1) forward speed (of the end-effector) and (2) planar pushing angle (*i.e.,* yaw orientation of the end-effector). The patch is not visualized.

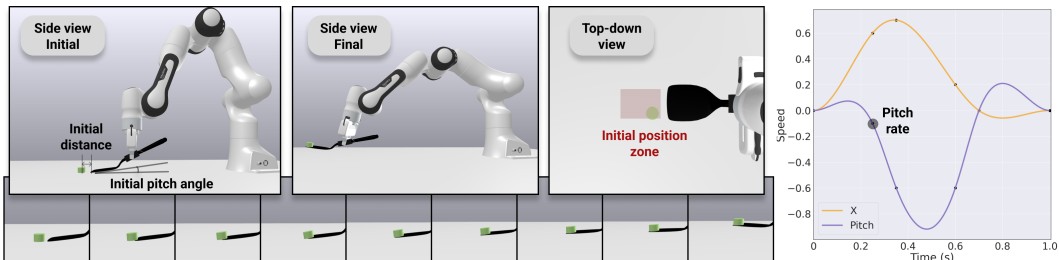

Figure A2: Visualization of the scooping trajectory and initial positions of the food piece in the Drake simulator. There are three action parameters: (1) initial distance (between the spatula and food piece), (2) initial pitch angle (of the spatula from the table), and (3) pitch rate (of the end-effector at time step $t=0.25$).

### A4.2 Dynamic scooping of food pieces

**Trajectory parameterization.** Here we detail the trajectory of the end-effector with the spatula designed for the task (Fig. A2). The end-effector velocity trajectory is generated using cubic spline with values clamped at five timesteps. The trajectory only varies in the X and pitch direction (in the world frame), while remaining zero in the other directions. The only value defining the trajectory that the task policy learns is the pitch rate, which is the pitch speed at the time $t=0.25$s and varies between $-0.2$rad/s and $0.2$rad/s. A positive pitch rate means the spatula lifting off the table late, while a negative one means lifting off early (see the effects in Fig. 8). The other two values that the task policy outputs are the initial pitch angle of the spatula from the table (varying from 2 to 10 degrees), and the initial distance between the spatula and the food piece (varying between 0.5cm to 2cm). Generally a higher initial pitch angle can help scoop under food pieces with flat bottom, and a smaller angle helps scoop under ellipsoidal shapes. We design such trajectories after extensive testing with food pieces of diverse geometric shapes and physical properties in both simulation and reality.

**Initial states.** The food piece is randomly placed in a box area of 8x6cm in front of the spatula; the initial distance is relative to the initial food piece location.

**Task policy parameterization.** The task policy is parameterized using a NAF again. In this task, it maps the initial 2D position of the food piece to the three action parameters: pitch rate, initial pitch angle, and initial distance.

**Hardware setup.** We use the commercially available OXO Nylon Square Turner[1] as the spatula used for scooping. It has a relatively thin edge (about 1.2mm) that helps scoop under thin pieces. A box-like, 3D-printed adapter with high-friction tape is mounted on the handle to help the parallel-jaw gripper grasp the spatula firmly. The exact 3D model of the spatula with the adapter is designed and used in the Drake simulator; the deformation effect as it bends against the table is not modeled in simulation.

---

[1]link: https://www.amazon.com/OXO-11107900LOW-Grips-Square-Turner/dp/B003L000SU

## A5  Additional details of experiments

### A5.1  Simulated adaptation

Table A2 shows the simulation parameters used in different simulated target environments for the three tasks (results shown in Table 3).

| Task | Parameter | WD | OOD-1 | OOD-2 | OOD-3 | OOD-4 | Range |
|------|-----------|-----|-------|-------|-------|-------|-------|
| | | | | Setting | | | |
| Pendulum | $m_1$ | 1.8 | 1.8 | **0.5** | 1.2 | **0.4** | $[1,2]$ |
| | $m_2$ | 1.2 | **0.3** | 1.8 | 1.8 | **2.6** | $[1,2]$ |
| | $b_1$ | 1.5 | 1.5 | 1.5 | **10.0** | 1.0 | $[1,2]$ |
| | $b_2$ | 1.5 | 1.5 | 1.5 | **10.0** | 2.0 | $[1,2]$ |
| Pushing | $\mu$ | 0.1 | **0.25** | 0.05 | 0.15 | **0.30** | $[0.05,0.2]$ |
| | $e$ | 1e5 | 5e4 | 1e5 | **5e6** | 1e5 | $[1e4,1e6]$ |
| | $\mu_p$ | 0.6 | **0.1** | **0.9** | **0.1** | **0.15** | $[0.2,0.8]$ |
| | $y_p$ | 0.05 | -0.1 | 0.05 | **-0.15** | 0.1 | $[-0.1,0.1]$ |
| Scooping | $\mu$ | 0.30 | **0.45** | **0.20** | 0.30 | 0.40 | $[0.25,0.4]$ |
| | $e$ | 5e4 | 1e4 | 5e4 | **1e6** | 1e5 | $[1e4,5e5]$ |
| | $g$ | 1 | 0 | 1 | 0 | **2** | $\{0,1\}$ |
| | $h$ | 2.0 | **1.4** | 2.2 | **2.8** | 1.9 | $[1.5,2.5]$ |

Table A2: Simulation parameters used in different simulated target environments for the three tasks. OOD parameters (outside the range used in adaptation policy training) are bolded. For $g$ in the scooping task, 0 stands for ellipsoid, 1 for cylinder, and 2 for box.

### A5.2  Real adaptation

In Fig. A7 and Fig. A9 we demonstrate additional visualizations of the pushing and scooping results with AdaptSim.

### A5.3  Additional studies

**Choice of the simulation parameter space.** To answer Q3, we perform a sensitivity analysis by fixing the target environment (OOD-1 in the double pendulum task) and varying the simulation parameter space. In OOD-1, the OOD parameter is $m_2 = 0.3$ while the range in $\Omega$ is $[1,2]$. Fig. A3 shows the results of reward achieved after adaptation for AdaptSim and the two Sys-ID baselines, as the range shifts further away from $m_2 = 0.3$ to $[1.1,2.1]$, $[1.2,2.2]$, and $[1.3,2.3]$. Sys-ID performance degrades rapidly, while AdaptSim is more robust.

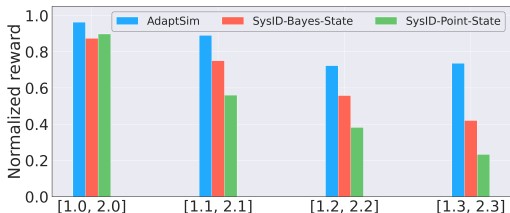

Figure A3: Adaptation results for AdaptSim and Sys-ID baselines in OOD-1 setting of the double pendulum task, with different $m_2$ ranges in $\Omega$ while $m_2 = 0.3$ in the target environment.

**Pitfalls of Sys-ID approaches.** Fig. A4 demonstrates the dynamics mismatch between simulation and reality, which illustrates the pitfall of SysID approaches. We plot a set of bottle trajectories from randomly sampled simulation parameters from $\Omega$ with a fixed robot action. We also plot the trajectories of Heavy bottle being pushed with the same action in reality. There are segments of real trajectories that are not well matched by the simulated ones, and a slight mismatch can lead to diverging final states (and hence different task rewards).

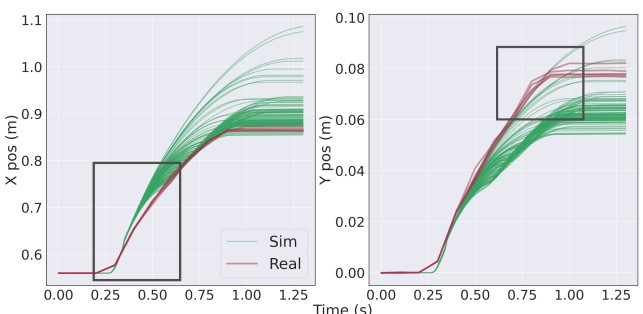

Figure A4: Comparison of trajectories from the simulation domain (green, simulated with randomly sampled simulation parameter settings) and from Heavy bottle in reality (red), with the same robot action applied. The real dynamics can be OOD from simulation (black boxes) while the final position of the bottle can be WD.

**Trade-off between real data budget and task performance convergance.** In Sec. 4.2 we introduce $N$, the number of initial simulation parameter distributions that are sampled at the beginning of Phase 2 and then adapt independently. There is a trade-off between the real data budget (linear to $N$) and convergence of task performance. Adapting more simulation parameter distributions simultaneously can potentially help the task performance converge faster but also require more real data. Fig. A5 shows the effect with the Light bottle in the pushing task. We vary $N$ from 1 to 4 — each simulation parameter distribution takes 2 trajectories at each iteration. $N=1$ shows slow and also worse asymptotic convergence, which shows that the parameter distribution can be trapped in a low-reward regime. $N=2$ performs the best with fastest convergence in terms of number of real trajectories used. Using higher $N$ shows slower convergence. Note that the convergence also depends on the dimension of the simulation parameter space $\Omega$ — we expect $N>2$ is needed for the best convergence rate once the dimension increases from 4 used in the pushing task.

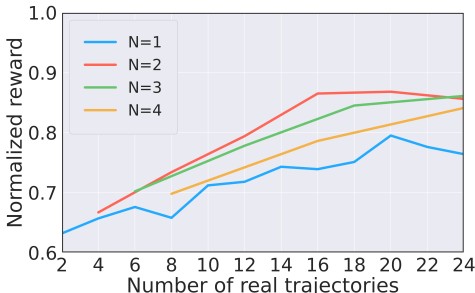

Figure A5: Task performance convergence with respect to the number of real trajectories used with varying $N$, the number of simulation parameter distributions adapting simultaneously in Phase 2 with the Light bottle in the pushing task.

**Sensitivity analysis on adaptation step size.** Adaption step size $\delta$ can affect the task performance convergence too — $\delta$ being too low can cause slow convergence, while $\delta$ being too high can prevent convergence since the simulation parameter distribution can "overshoot" the optimal one by a large margin. Fig. A6 shows the effect of adaptation step size ranging from 0.05 to 0.20 in OOD-1 setting of the double pendulum task. $\delta = 0.10$ performs the best while $\delta = 0.05$ shows slower convergence. $\delta = 0.15$ also achieves similar asymptotic performance but the reward is less unstable during adaptation, while with $\delta = 0.20$ the reward does not converge at all.

**Comparison of simulation runtime.** Compared to Sys-ID baselines, AdaptSim requires significantly longer simulation runtime for training the adaptation policy in Phase 1. For example: SysID-Bayes uses roughly 6 hours of simulation walltime to perform 10 iterations of adaptation in the scooping task while AdaptSim would take 36 hours for Phase 1, and 30 minutes for Phase 2 (i.e., 3 minutes per iteration), using the same computation setup. However, we re-use the same adaptation policy for different food pieces in the scooping task, which amortizes the simulation cost.

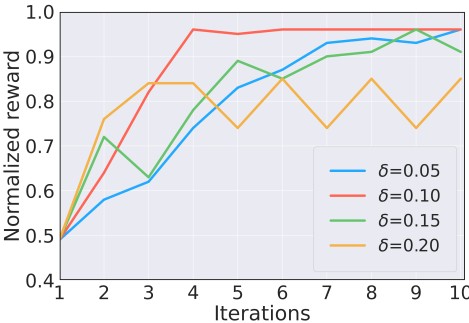

Figure A6: Normalized reward at each adaptation iteration using different adaptation step size $\delta$, in OOD-1 setting of the pendulum task.

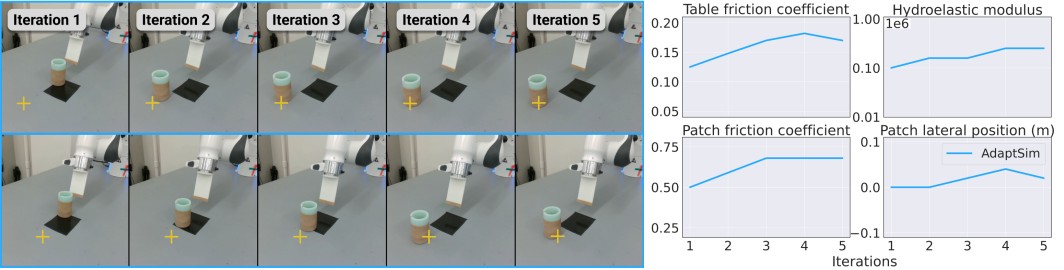

Figure A7: Adaptation results of the pushing task with two different target locations (yellow cross, top and bottom rows) over iterations. The right figure shows the inferred simulation parameter distribution (mean only).

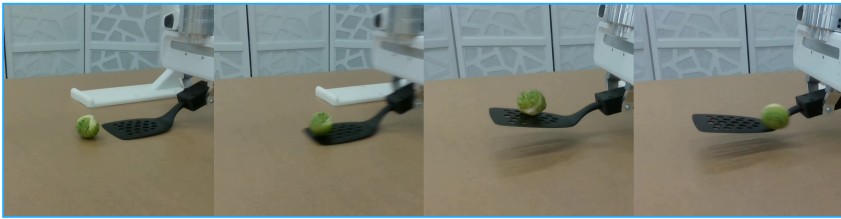

Figure A8: AdaptSim fails to synthesize a task policy for scooping up Brussels sprout. We consider such environment extremely OOD from the simulation domain.

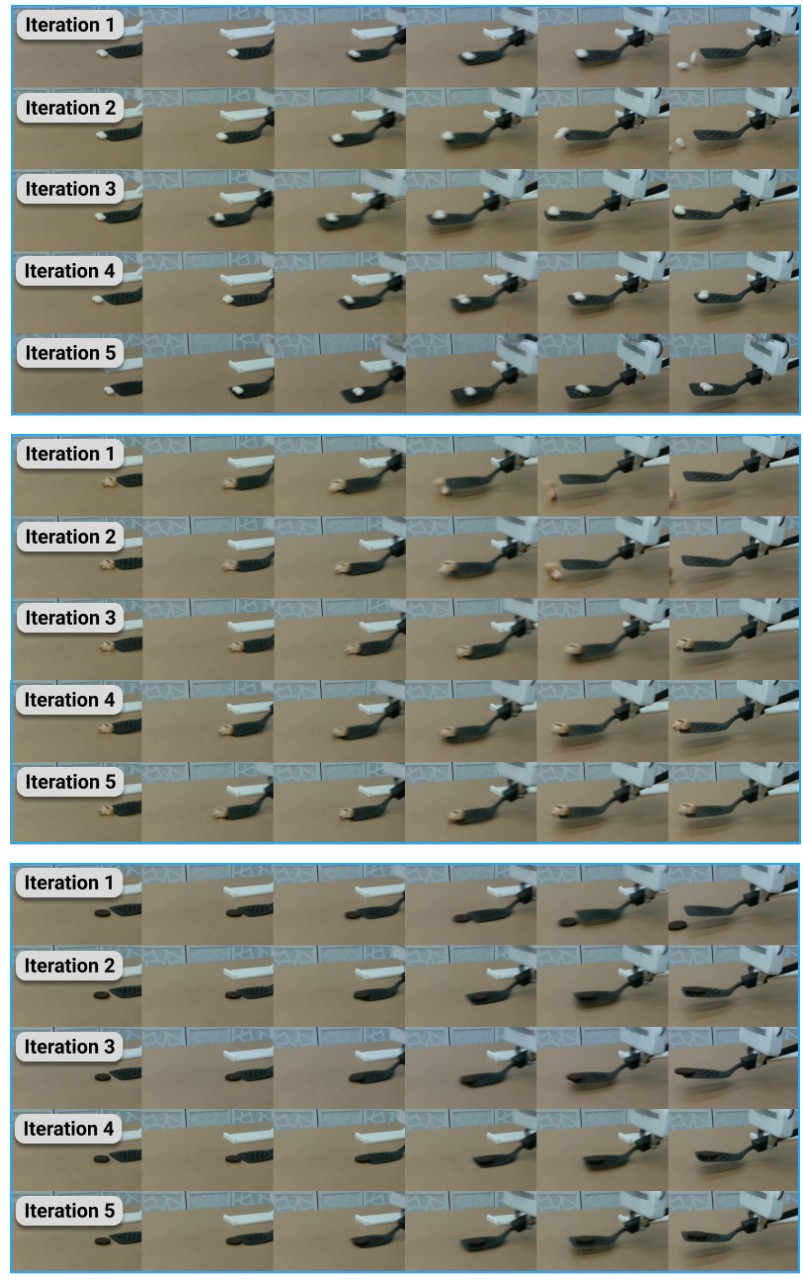

Figure A9: Adaptation results of scooping up (top) chocolate raisins, (middle) mushroom slice, and (bottom) Oreo cookie with AdaptSim.

