# OpenReview forum: "AdaptSim: Task-Driven Simulation Adaptation for Sim-to-Real Transfer"
_robot-learning.org/CoRL/2023/Workshop/TGR — CoRL 2023 Workshop TGR Poster_

### Official Review · Reviewer_asLt · 2023-10-19

**Rating:** 8
**Confidence:** 5

**Review:**

The method for bridging sim-to-real gap is well justified and could be helpful in constructing simulation environments for training real-world deployable skills.

---

### Official Review · Reviewer_2EhP · 2023-10-19
**Strong accept**

**Rating:** 9
**Confidence:** 4

**Review:**

Great contribution in using the adaptative simulator for assisting sim2real policy transfer. Good real-world results in contact-rich tasks.

---

### Decision · Program_Chairs · 2023-10-20

**Decision:**

Accept (Poster)

**Comment:**

Great paper and closely aligned topic!